# Differentiable MPC for End-to-end Planning and Control

**Brandon Amos**[1]   **Ivan Dario Jimenez Rodriguez**[2]   **Jacob Sacks**[2]
**Byron Boots**[2]   **J. Zico Kolter**[13]
[1]Carnegie Mellon University   [2]Georgia Tech   [3]Bosch Center for AI

## Abstract

We present foundations for using Model Predictive Control (MPC) as a differentiable policy class for reinforcement learning in continuous state and action spaces. This provides one way of leveraging and combining the advantages of model-free and model-based approaches. Specifically, we differentiate through MPC by using the KKT conditions of the convex approximation at a fixed point of the controller. Using this strategy, we are able to learn the cost and dynamics of a controller via end-to-end learning. Our experiments focus on imitation learning in the pendulum and cartpole domains, where we learn the cost and dynamics terms of an MPC policy class. We show that our MPC policies are significantly more data-efficient than a generic neural network and that our method is superior to traditional system identification in a setting where the expert is unrealizable.

## 1   Introduction

Model-free reinforcement learning has achieved state-of-the-art results in many challenging domains. However, these methods learn black-box control policies and typically suffer from poor sample complexity and generalization. Alternatively, model-based approaches seek to model the environment the agent is interacting in. Many model-based approaches utilize Model Predictive Control (MPC) to perform complex control tasks [González et al., 2011, Lenz et al., 2015, Liniger et al., 2014, Kamel et al., 2015, Erez et al., 2012, Alexis et al., 2011, Bouffard et al., 2012, Neunert et al., 2016]. MPC leverages a predictive model of the controlled system and solves an optimization problem online in a receding horizon fashion to produce a sequence of control actions. Usually the first control action is applied to the system, after which the optimization problem is solved again for the next time step.

Formally, MPC requires that at each time step we solve the optimization problem:

$$\operatorname*{argmin}_{x_{1:T} \in \mathcal{X}, u_{1:T} \in \mathcal{U}} \sum_{t=1}^{T} C_t(x_t, u_t) \ \text{ subject to } \ x_{t+1} = f(x_t, u_t), \ \ x_1 = x_{\text{init}}, \tag{1}$$

where $x_t, u_t$ are the state and control at time $t$, $\mathcal{X}$ and $\mathcal{U}$ are constraints on valid states and controls, $C_t : \mathcal{X} \times \mathcal{U} \to \mathbb{R}$ is a (potentially time-varying) cost function, $f : \mathcal{X} \times \mathcal{U} \to \mathcal{X}$ is a dynamics model, and $x_{\text{init}}$ is the initial state of the system. The optimization problem in Equation (1) can be efficiently solved in many ways, for example with the finite-horizon iterative Linear Quadratic Regulator (iLQR) algorithm [Li and Todorov, 2004]. Although these techniques are widely used in control domains, much work in deep reinforcement learning or imitation learning opts instead to use a much simpler policy class such as a linear function or neural network. The advantages of these policy classes is that they are differentiable and the loss can be directly optimized with respect to them while it is typically not possible to do full end-to-end learning with model-based approaches.

In this paper, we consider the task of learning MPC-based policies in an end-to-end fashion, illustrated in Figure 1. That is, we treat MPC as a generic policy class $u = \pi(x_{\text{init}}; C, f)$ parameterized by some representations of the cost $C$ and dynamics model $f$. By differentiating *through* the optimization problem, we can learn the costs and dynamics model to perform a desired task. This is in contrast to

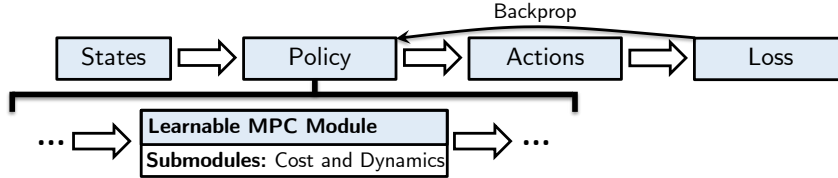

Figure 1: **Illustration of our contribution:** A learnable MPC module that can be integrated into a larger end-to-end reinforcement learning pipeline. Our method allows the controller to be updated with gradient information directly from the task loss.

regressing on collected dynamics or trajectory rollout data and learning each component in isolation, and comes with the typical advantages of end-to-end learning (the ability to train directly based upon the task loss of interest, the ability to "specialize" parameter for a given task, etc).

Still, efficiently differentiating through a complex policy class like MPC is challenging. Previous work with similar aims has either simply unrolled and differentiated through a simple optimization procedure [Tamar et al., 2017] or has considered generic optimization solvers that do not scale to the size of MPC problems [Amos and Kolter, 2017]. This paper makes the following two contributions to this space. First, we provide an efficient method for *analytically* differentiating through an iterative non-convex optimization procedure based upon a box-constrained iterative LQR solver [Tassa et al., 2014]; in particular, we show that the analytical derivative can be computed using *one additional* backward pass of a modified iterative LQR solver. Second, we empirically show that in imitation learning scenarios we can recover the *cost* and *dynamics* from an MPC expert with a loss based only on the actions (and not states). In one notable experiment, we show that directly optimizing the imitation loss results in better performance than vanilla system identification.

## 2   Background and Related Work

**Pure model-free techniques for policy search** have demonstrated promising results in many domains by learning *reactive polices* which directly map observations to actions [Mnih et al., 2013, Oh et al., 2016, Gu et al., 2016b, Lillicrap et al., 2015, Schulman et al., 2015, 2016, Gu et al., 2016a]. Despite their success, model-free methods have many drawbacks and limitations, including a lack of interpretability, poor generalization, and a high sample complexity. **Model-based methods** are known to be more sample-efficient than their model-free counterparts. These methods generally rely on learning a dynamics model directly from interactions with the real system and then integrate the learned model into the control policy [Schneider, 1997, Abbeel et al., 2006, Deisenroth and Rasmussen, 2011, Heess et al., 2015, Boedecker et al., 2014]. More recent approaches use a deep network to learn low-dimensional latent state representations and associated dynamics models in this learned representation. They then apply standard trajectory optimization methods on these learned embeddings [Lenz et al., 2015, Watter et al., 2015, Levine et al., 2016]. However, these methods still require a manually specified and hand-tuned cost function, which can become even more difficult in a latent representation. Moreover, there is no guarantee that the learned dynamics model can accurately capture portions of the state space relevant for the task at hand.

To leverage the benefits of both approaches, there has been significant interest in **combining the model-based and model-free paradigms.** In particular, much attention has been dedicated to utilizing model-based priors to accelerate the model-free learning process. For instance, synthetic training data can be generated by model-based control algorithms to guide the policy search or prime a model-free policy [Sutton, 1990, Theodorou et al., 2010, Levine and Abbeel, 2014, Gu et al., 2016b, Venkatraman et al., 2016, Levine et al., 2016, Chebotar et al., 2017, Nagabandi et al., 2017, Sun et al., 2017]. [Bansal et al., 2017] learns a controller and then distills it to a neural network policy which is then fine-tuned with model-free policy learning. However, this line of work usually keeps the model separate from the learned policy.

Alternatively, the policy can include an **explicit planning module** which *leverages learned models* of the system or environment, both of which are learned through model-free techniques. For example, the classic Dyna-Q algorithm [Sutton, 1990] simultaneously learns a model of the environment and uses it to plan. More recent work has explored incorporating such structure into deep networks and learning the policies in an end-to-end fashion. Tamar et al. [2016] uses a recurrent network to predict

the value function by approximating the value iteration algorithm with convolutional layers. Karkus et al. [2017] connects a dynamics model to a planning algorithm and formulates the policy as a structured recurrent network. Silver et al. [2016] and Oh et al. [2017] perform multiple rollouts using an abstract dynamics model to predict the value function. A similar approach is taken by Weber et al. [2017] but directly predicts the next action and reward from rollouts of an explicit environment model. Farquhar et al. [2017] extends model-free approaches, such as DQN [Mnih et al., 2015] and A3C [Mnih et al., 2016], by planning with a tree-structured neural network to predict the cost-to-go. While these approaches have demonstrated impressive results in discrete state and action spaces, they are not applicable to continuous control problems.

To tackle continuous state and action spaces, Pascanu et al. [2017] propose a neural architecture which uses an abstract environmental model to plan and is trained directly from an external task loss. Pong et al. [2018] learn goal-conditioned value functions and use them to plan single or multiple steps of actions in an MPC fashion. Similarly, Pathak et al. [2018] train a goal-conditioned policy to perform rollouts in an abstract feature space but ground the policy with a loss term which corresponds to true dynamics data. The aforementioned approaches can be interpreted as a distilled optimal controller which does not separate components for the cost and dynamics. Taking this analogy further, another strategy is to differentiate through an optimal control algorithm itself. Okada et al. [2017] and Pereira et al. [2018] present a way to differentiate through path integral optimal control [Williams et al., 2016, 2017] and learn a planning policy end-to-end. Srinivas et al. [2018] shows how to embed differentiable planning (unrolled gradient descent over actions) within a goal-directed policy. In a similar vein, Tamar et al. [2017] differentiates through an iterative LQR (iLQR) solver [Li and Todorov, 2004, Xie et al., 2017, Tassa et al., 2014] to learn a cost-shaping term offline. This shaping term enables a shorter horizon controller to approximate the behavior of a solver with a longer horizon to save computation during runtime.

**Contributions of our paper.** All of these methods require differentiating through planning procedures by explicitly "unrolling" the optimization algorithm itself. While this is a reasonable strategy, it is both memory- and computationally-expensive and challenging when unrolling through many iterations because the time- and space-complexity of the backward pass grows linearly with the forward pass. In contrast, we address this issue by showing how to *analytically* differentiate through the fixed point of a nonlinear MPC solver. Specifically, we compute the derivatives of an iLQR solver with a *single* LQR step in the backward pass. This makes the learning process more computationally tractable while still allowing us to plan in continuous state and action spaces. Unlike model-free approaches, explicit cost and dynamics components can be extracted and analyzed on their own. Moreover, in contrast to pure model-based approaches, the dynamics model and cost function can be learned entirely end-to-end.

## 3 Differentiable LQR

Discrete-time finite-horizon LQR is a well-studied control method that optimizes a convex quadratic objective function with respect to affine state-transition dynamics from an initial system state $x_{\text{init}}$. Specifically, LQR finds the optimal nominal trajectory $\tau_{1:T}^\star = \{x_t, u_t\}_{1:T}$ by solving the optimization problem

$$\tau_{1:T}^\star = \operatorname*{argmin}_{\tau_{1:T}} \ \sum_t \frac{1}{2}\tau_t^\top C_t \tau_t + c_t^\top \tau_t \ \ \text{subject to} \ \ x_1 = x_{\text{init}}, \ x_{t+1} = F_t \tau_t + f_t. \qquad (2)$$

From a policy learning perspective, this can be interpreted as a module with unknown parameters $\theta = \{C, c, F, f\}$, which can be integrated into a larger end-to-end learning system. The learning process involves taking derivatives of some loss function $\ell$, which are then used to update the parameters. Instead of directly computing each of the individual gradients, we present an efficient way of computing the derivatives of the loss function with respect to the parameters

$$\frac{\partial \ell}{\partial \theta} = \frac{\partial \ell}{\partial \tau_{1:T}^\star} \frac{\partial \tau_{1:T}^\star}{\partial \theta}. \qquad (3)$$

By interpreting LQR from an optimization perspective [Boyd, 2008], we associate dual variables $\lambda_{1:T}$ with the state constraints. The Lagrangian of the optimization problem is then given by

$$\mathcal{L}(\tau, \lambda) = \sum_t \frac{1}{2}\tau_t^\top C_t \tau_t + \sum_{t=0}^{T-1} \lambda_t^\top (F_t \tau_t + f_t - x_{t+1}), \qquad (4)$$

**Module 1** Differentiable LQR                                            *(The LQR algorithm is defined in Appendix A)*

**Input:** Initial state $x_{\text{init}}$
**Parameters:** $\theta = \{C, c, F, f\}$

**Forward Pass:**
1: $\tau^\star_{1:T} = \text{LQR}_T(x_{\text{init}}; C, c, F, f)$                                               ▷ Solve (2)
2: Compute $\lambda^\star_{1:T}$ with (7)

**Backward Pass:**
1: $d^\star_{\tau_{1:T}} = \text{LQR}_T(0; C, \nabla_{\tau^\star}\ell, F, 0)$    ▷ Solve (9), ideally reusing the factorizations from the forward pass
2: Compute $d^\star_{\lambda_{1:T}}$ with (7)
3: Compute the derivatives of $\ell$ with respect to $C$, $c$, $F$, $f$, and $x_{\text{init}}$ with (8)

---

where the initial constraint $x_1 = x_{\text{init}}$ is represented by setting $F_0 = 0$ and $f_0 = x_{\text{init}}$. Differentiating Equation (4) with respect to $\tau^\star_t$ yields

$$\nabla_{\tau_t}\mathcal{L}(\tau^\star, \lambda^\star) = C_t\tau^\star_t + C_t + F_t^\top\lambda^\star_t - \begin{bmatrix} \lambda^\star_{t-1} \\ 0 \end{bmatrix} = 0, \tag{5}$$

Thus, the normal approach to solving LQR problems with dynamic Riccati recursion can be viewed as an efficient way of solving the KKT system

$$\overbrace{\begin{bmatrix} \ddots & & & & & \\ & C_t & F_t^\top & & & \\ & F_t & & [-I \quad 0] & & \\ & & \begin{bmatrix} -I \\ 0 \end{bmatrix} & C_{t+1} & F_{t+1}^\top & \\ & & & F_{t+1} & & \\ & & & & & \ddots \end{bmatrix}}^{K} \begin{bmatrix} \vdots \\ \tau^\star_t \\ \lambda^\star_t \\ \tau^\star_{t+1} \\ \lambda^\star_{t+1} \\ \vdots \end{bmatrix} = - \begin{bmatrix} \vdots \\ c_t \\ f_t \\ c_{t+1} \\ f_{t+1} \\ \vdots \end{bmatrix}. \tag{6}$$

with column labels $\tau_t \quad \lambda_t \quad \tau_{t+1} \quad \lambda_{t+1}$.

Given an optimal nominal trajectory $\tau^\star_{1:T}$, Equation (5) shows how to compute the optimal dual variables $\lambda$ with the backward recursion

$$\lambda^\star_T = C_{T,x}\tau^\star_T + c_{T,x} \qquad \lambda^\star_t = F_{t,x}^\top\lambda^\star_{t+1} + C_{t,x}\tau^\star_t + c_{t,x}, \tag{7}$$

where $C_{t,x}$, $c_{t,x}$, and $F_{t,x}$ are the first block-rows of $C_t$, $c_t$, and $F_t$, respectively. Now that we have the optimal trajectory and dual variables, we can compute the gradients of the loss with respect to the parameters. Since LQR is a constrained convex quadratic $\text{argmin}$, the derivatives of the loss with respect to the LQR parameters can be obtained by implicitly differentiating the KKT conditions. Applying the approach from Section 3 of Amos and Kolter [2017], the derivatives are

$$\nabla_{C_t}\ell = \frac{1}{2}\left(d^\star_{\tau_t} \otimes \tau^\star_t + \tau^\star_t \otimes d^\star_{\tau_t}\right) \qquad \nabla_{c_t}\ell = d^\star_{\tau_t} \qquad \nabla_{x_{\text{init}}}\ell = d^\star_{\lambda_0}$$
$$\nabla_{F_t}\ell = d^\star_{\lambda_{t+1}} \otimes \tau^\star_t + \lambda^\star_{t+1} \otimes d^\star_{\tau_t} \qquad \nabla_{f_t}\ell = d^\star_{\lambda_t} \tag{8}$$

where $\otimes$ is the outer product operator, and $d^\star_\tau$ and $d^\star_\lambda$ are obtained by solving the linear system

$$K \begin{bmatrix} \vdots \\ d^\star_{\tau_t} \\ d^\star_{\lambda_t} \\ \vdots \end{bmatrix} = - \begin{bmatrix} \vdots \\ \nabla_{\tau^\star_t}\ell \\ 0 \\ \vdots \end{bmatrix}. \tag{9}$$

We observe that Equation (9) is of the same form as the linear system in Equation (6) for the LQR problem. Therefore, we can leverage this insight and solve Equation (9) efficiently by solving another LQR problem that replaces $c_t$ with $\nabla_{\tau^\star_t}\ell$ and $f_t$ with 0. Moreover, this approach enables us to re-use the factorization of $K$ from the forward pass instead of recomputing. Module 1 summarizes the forward and backward passes for a differentiable LQR module.

# 4 Differentiable MPC

While LQR is a powerful tool, it does not cover realistic control problems with non-linear dynamics and cost. Furthermore, most control problems have natural bounds on the control space that can often be expressed as box constraints. These highly non-convex problems, which we will refer to as model predictive control (MPC), are well-studied in the control literature and can be expressed in the general form

$$\tau_{1:T}^{\star} = \underset{\tau_{1:T}}{\operatorname{argmin}} \ \sum_t C_{\theta,t}(\tau_t) \ \text{ subject to } \ x_1 = x_{\text{init}}, \ x_{t+1} = f_\theta(\tau_t), \ \underline{u} \le u \le \overline{u}, \qquad (10)$$

where the non-convex cost function $C_\theta$ and non-convex dynamics function $f_\theta$ are (potentially) parameterized by some $\theta$. We note that more generic constraints on the control and state space can be represented as penalties and barriers in the cost function. The standard way of solving the control problem Equation (10) is by iteratively forming and optimizing a convex approximation

$$\tau_{1:T}^{i} = \underset{\tau_{1:T}}{\operatorname{argmin}} \ \sum_t \tilde{C}_{\theta,t}^{i}(\tau_t) \ \text{ subject to } \ x_1 = x_{\text{init}}, \ x_{t+1} = \tilde{f}_\theta^i(\tau_t), \ \underline{u} \le u \le \overline{u}, \qquad (11)$$

where we have defined the second-order Taylor approximation of the cost around $\tau^i$ as

$$\tilde{C}_{\theta,t}^{i} = C_{\theta,t}(\tau_t^i) + (p_t^i)^\top (\tau_t - \tau_t^i) + \frac{1}{2}(\tau_t - \tau_t^i)^\top H_t^i (\tau_t - \tau_t^i) \qquad (12)$$

with $p_t^i = \nabla_{\tau_t^i} C_{\theta,t}$ and $H_t^i = \nabla_{\tau_t^i}^2 C_{\theta,t}$. We also have a first-order Taylor approximation of the dynamics around $\tau^i$ as

$$\tilde{f}_{\theta,t}^{i}(\tau_t) = f_{\theta,t}(\tau_t^i) + F_t^i(\tau_t - \tau_t^i) \qquad (13)$$

with $F_t^i = \nabla_{\tau_t^i} f_{\theta,t}$. In practice, a fixed point of Equation (11) is often reached, especially when the dynamics are smooth. As such, differentiating the non-convex problem Equation (10) can be done exactly by using the final convex approximation. Without the box constraints, the fixed point in Equation (11) could be differentiated with LQR as we show in Section 3. In the next section, we will show how to extend this to the case where we have box constraints on the controls as well.

## 4.1 Differentiating Box-Constrained QPs

First, we consider how to differentiate a more generic box-constrained convex QP of the form

$$x^{\star} = \underset{x}{\operatorname{argmin}} \ \frac{1}{2}x^\top Q x + p^\top x \ \text{ subject to } \ Ax = b, \ \underline{x} \le x \le \overline{x}. \qquad (14)$$

Given active inequality constraints at the solution in the form $\tilde{G}x = \tilde{h}$, this problem turns into an equality-constrained optimization problem with the solution given by the linear system

$$\begin{bmatrix} Q & A^\top & \tilde{G}^\top \\ A & 0 & 0 \\ \tilde{G} & 0 & 0 \end{bmatrix} \begin{bmatrix} x^\star \\ \lambda^\star \\ \tilde{\nu}^\star \end{bmatrix} = - \begin{bmatrix} p \\ b \\ \tilde{h} \end{bmatrix} \qquad (15)$$

With some loss function $\ell$ that depends on $x^\star$, we can use the approach in Amos and Kolter [2017] to obtain the derivatives of $\ell$ with respect to $Q$, $p$, $A$, and $b$ as

$$\nabla_Q \ell = \frac{1}{2}(d_x^\star \otimes x^\star + x^\star \otimes d_x^\star) \qquad \nabla_p \ell = d_x^\star \qquad \nabla_A \ell = d_\lambda^\star \otimes x^\star + \lambda^\star \otimes d_x^\star \qquad \nabla_b \ell = -d_\lambda^\star \quad (16)$$

where $d_x^\star$ and $d_\lambda^\star$ are obtained by solving the linear system

$$\begin{bmatrix} Q & A^\top & \tilde{G}^\top \\ A & 0 & 0 \\ \tilde{G} & 0 & 0 \end{bmatrix} \begin{bmatrix} d_x^\star \\ d_\lambda^\star \\ d_{\tilde{\nu}}^\star \end{bmatrix} = - \begin{bmatrix} \nabla_{x^\star}\ell \\ 0 \\ 0 \end{bmatrix} \qquad (17)$$

The constraint $\tilde{G}d_x^\star = 0$ is equivalent to the constraint $d_{x_i}^\star = 0$ if $x_i^\star \in \{\underline{x}_i, \overline{x}_i\}$. Thus solving the system in Equation (17) is equivalent to solving the optimization problem

$$d_x^\star = \underset{d_x}{\operatorname{argmin}} \ \frac{1}{2}d_x^\top Q d_x + (\nabla_{x^\star}\ell)^\top d_x \ \text{ subject to } \ Ad_x = 0, \ d_{x_i} = 0 \ \text{ if } \ x_i^\star \in \{\underline{x}_i, \overline{x}_i\} \quad (18)$$

**Module 2** Differentiable MPC                    *(The MPC algorithm is defined in Appendix A)*

**Given:** Initial state $x_{\text{init}}$ and initial control sequence $u_{\text{init}}$
**Parameters:** $\theta$ of the objective $C_\theta(\tau)$ and dynamics $f_\theta(\tau)$

**Forward Pass:**
1: $\tau_{1:T}^\star = \text{MPC}_{T,\underline{u},\overline{u}}(x_{\text{init}}, u_{\text{init}}; C_\theta, F_\theta)$                    ▷ Solve Equation (10)
2: *The solver should reach the fixed point in* (11) *to obtain approximations to the cost $H_\theta^n$ and dynamics $F_\theta^n$*
3: Compute $\lambda_{1:T}^\star$ with (7)

**Backward Pass:**
1: $\tilde{F}_\theta^n$ is $F_\theta^n$ with the rows corresponding to the tight control constraints zeroed
2: $d_{\tau_{1:T}}^\star = \text{LQR}_T(0; H_\theta^n, \nabla_{\tau^\star}\ell, \tilde{F}_\theta^n, 0)$ ▷ Solve (19), ideally reusing the factorizations from the forward pass
3: Compute $d_{\lambda_{1:T}}^\star$ with (7)
4: Differentiate $\ell$ with respect to the approximations $H_\theta^n$ and $F_\theta^n$ with (8)
5: Differentiate these approximations with respect to $\theta$ and use the chain rule to obtain $\partial\ell/\partial\theta$

## 4.2 Differentiating MPC with Box Constraints

At a fixed point, we can use Equation (16) to compute the derivatives of the MPC problem, where $d_\tau^\star$ and $d_\lambda^\star$ are found by solving the linear system in Equation (9) with the additional constraint that $d_{u_{t,i}} = 0$ if $u_{t,i}^\star \in \{\underline{u}_{t,i}, \overline{u}_{t,i}\}$. Solving this system can be equivalently written as a zero-constrained LQR problem of the form

$$d_{\tau_{1:T}}^\star = \underset{d_{\tau_{1:T}}}{\text{argmin}} \quad \sum_t \frac{1}{2} d_{\tau_t}^\top H_t^n d_{\tau_t} + (\nabla_{\tau_t^\star}\ell)^\top d_{\tau_t}$$
$$\text{subject to} \ \ d_{x_1} = 0, \ d_{x_{t+1}} = F_t^n d_{\tau_t}, \ d_{u_{t,i}} = 0 \ \text{ if } \ u_i^\star \in \{\underline{u}_{t,i}, \overline{u}_{t,i}\} \tag{19}$$

where $n$ is the iteration that Equation (11) reaches a fixed point, and $H^n$ and $F^n$ are the corresponding approximations to the objective and dynamics defined earlier. Module 2 summarizes the proposed differentiable MPC module. To solve the MPC problem in Equation (10) and reach the fixed point in Equation (11), we use the box-DDP heuristic [Tassa et al., 2014]. For the zero-constrained LQR problem in Equation (19) to compute the derivatives, we use an LQR solver that zeros the appropriate controls.

## 4.3 Drawbacks of Our Approach

Sometimes the controller does not run for long enough to reach a fixed point of Equation (11), or a fixed point doesn't exist, which often happens when using neural networks to approximate the dynamics. When this happens, Equation (19) cannot be used to differentiate through the controller, because it assumes a fixed point. Differentiating through the final iLQR iterate that's not a fixed point will usually give the wrong gradients. Treating the iLQR procedure as a compute graph and differentiating through the unrolled operations is a reasonable alternative in this scenario that obtains surrogate gradients to the control problem. However, as we empirically show in Section 5.1, the backward pass of this method scales linearly with the number of iLQR iterations used in the forward. Instead, fixed-point differentiation is constant time and only requires a single iLQR solve.

# 5 Experimental Results

In this section, we present several results that highlight the performance and capabilities of differentiable MPC in comparison to neural network policies and vanilla system identification (SysId). We show 1) superior runtime performance compared to an unrolled solver, 2) the ability of our method to recover the cost and dynamics of a controller with imitation, and 3) the benefit of directly optimizing the task loss over vanilla SysId.

We have released our differentiable MPC solver as a standalone open source package that is available at https://github.com/locuslab/mpc.pytorch and our experimental code for this paper is also openly available at https://github.com/locuslab/differentiable-mpc. Our experiments are implemented with PyTorch [Paszke et al., 2017].

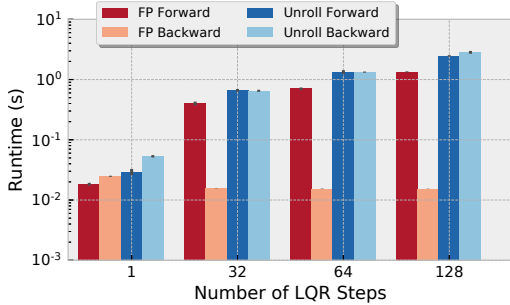

Figure 2: Runtime comparison of fixed point differentiation (FP) to unrolling the iLQR solver (Unroll), averaged over 10 trials.

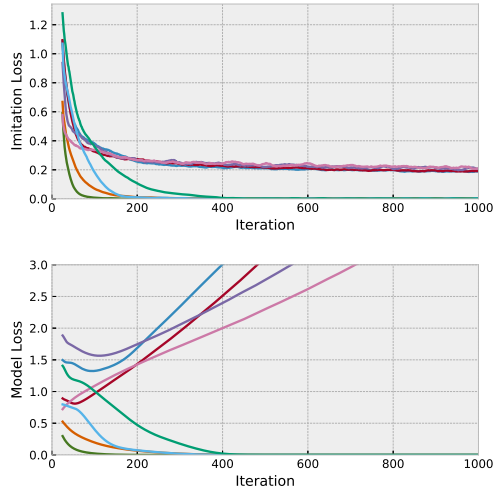

Figure 3: Model and imitation losses for the LQR imitation learning experiments.

## 5.1 MPC Solver Performance

Figure 2 highlights the performance of our differentiable MPC solver. We compare to an alternative version where each box-constrained iLQR iteration is individually unrolled, and gradients are computed by differentiating through the entire unrolled chain. As illustrated in the figure, these unrolled operations incur a substantial extra cost. Our differentiable MPC solver 1) is slightly more computationally efficient even in the forward pass, as it does not need to create and maintain the backward pass variables; 2) is more memory efficient in the forward pass for this same reason (by a factor of the number of iLQR iterations); and 3) is *significantly* more efficient in the backward pass, especially when a large number of iLQR iterations are needed. The backward pass is essentially free, as it can reuse all the factorizations for the forward pass and does not require multiple iterations.

## 5.2 Imitation Learning: Linear-Dynamics Quadratic-Cost (LQR)

In this section, we show results to validate the MPC solver and gradient-based learning approach for an imitation learning problem. The expert and learner are LQR controllers that share all information except for the linear system dynamics $f(x_t, u_t) = Ax_t + Bu_t$. The controllers have the same quadratic cost (the identity), control bounds $[-1, 1]$, horizon (5 timesteps), and 3-dimensional state and control spaces. Though the dynamics can also be recovered by fitting next-state transitions, we show that we can alternatively use imitation learning to recover the dynamics using only controls.

Given an initial state $x$, we can obtain nominal actions from the controllers as $u_{1:T}(x; \theta)$, where $\theta = \{A, B\}$. We randomly initialize the learner's dynamics with $\hat{\theta}$ and minimize the **imitation loss**

$$\mathcal{L} = \mathbb{E}_x \left[ ||\tau_{1:T}(x; \theta) - \tau_{1:T}(x; \hat{\theta})||_2^2 \right], .$$

We do learning by differentiating $\mathcal{L}$ with respect to $\hat{\theta}$ (using mini-batches with 32 examples) and taking gradient steps with RMSprop [Tieleman and Hinton, 2012]. Figure 3 shows the model and imitation loss of eight randomly sampled initial dynamics, where the **model loss** is $\mathrm{MSE}(\theta, \hat{\theta})$. The model converges to the true parameters in half of the trials and achieves a perfect imitation loss. The other trials get stuck in a local minimum of the imitation loss and causes the approximate model to significantly diverge from the true model. These faulty trials highlight that despite the LQR problem being convex, the optimization problem of some loss function w.r.t. the controller's parameters is a (potentially difficult) non-convex optimization problem that typically does not have convergence guarantees.

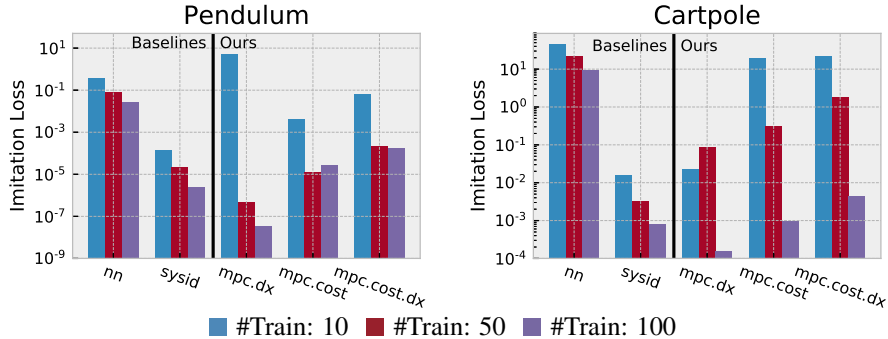

Figure 4: Learning results on the (simple) pendulum and cartpole environments. We select the best validation loss observed during the training run and report the best test loss.

## 5.3 Imitation Learning: Non-Convex Continuous Control

We next demonstrate the ability of our method to do imitation learning in the pendulum and cartpole benchmark domains. Despite being simple tasks, they are relatively challenging for a generic poicy to learn quickly in the imitation learning setting. In our experiments we use MPC experts and learners that produce a nominal action sequence $u_{1:T}(x; \theta)$ where $\theta$ parameterizes the model that's being optimized. The goal of these experiments is to optimize the imitation loss $\mathcal{L} = \mathbb{E}_x \left[ ||u_{1:T}(x; \theta) - u_{1:T}(x; \hat{\theta})||_2^2 \right]$, again which we can uniquely do using *only* observed controls and *no* observations. We consider the following methods:

**Baselines:** *nn* is an LSTM that takes the state $x$ as input and predicts the nominal action sequence. In this setting we optimize the imitation loss directly. *sysid* assumes the cost of the controller is known and approximates the parameters of the dynamics by optimizing the next-state transitions.

**Our Methods:** *mpc.dx* assumes the cost of the controller is known and approximates the parameters of the dynamics by directly optimizing the imitation loss. *mpc.cost* assumes the dynamics of the controller is known and approximates the cost by directly optimizing the imitation loss. *mpc.cost.dx* approximates both the cost and parameters of the dynamics of the controller by directly optimizing the imitation loss.

In all settings that involve learning the dynamics (*sysid*, *mpc.dx*, and *mpc.cost.dx*) we use a parameterized version of the true dynamics. In the pendulum domain, the parameters are the mass, length, and gravity; and in the cartpole domain, the parameters are the cart's mass, pole's mass, gravity, and length. For cost learning in *mpc.cost* and *mpc.cost.dx* we parameterize the cost of the controller as the weighted distance to a goal state $C(\tau) = ||w_g \circ (\tau - \tau_g)||_2^2$. We have found that simultaneously learning the weights $w_g$ and goal state $\tau_g$ is instable and in our experiments we alternate learning of $w_g$ and $\tau_g$ independently every 10 epochs. We collected a dataset of trajectories from an expert controller and vary the number of trajectories our models are trained on. A single trial of our experiments takes 1-2 hours on a modern CPU. We optimize the *nn* setting with Adam [Kingma and Ba, 2014] with a learning rate of $10^{-4}$ and all other settings are optimized with RMSprop [Tieleman and Hinton, 2012] with a learning rate of $10^{-2}$ and a decay term of $0.5$.

Figure 4 shows that in nearly every case we are able to directly optimize the imitation loss with respect to the controller and we significantly outperform a general neural network policy trained on the same information. In many cases we are able to recover the true cost function and dynamics of the expert. More information about the training and validation losses are in Appendix B. The comparison between our approach *mpc.dx* and SysId is notable, as we are able to recover equivalent performance to SysId with our models using *only* the control information and *without* using state information.

Again, while we emphasize that these are simple tasks, there are stark differences between the approaches. Unlike the generic network-based imitation learning, the MPC policy can exploit its inherent structure. Specifically, because the network contains a well-defined notion of the dynamics and cost, it is able to learn with much lower sample complexity that a typical network. But unlike pure system identification (which would be reasonable only for the case where the physical parameters are unknown but all other costs are known), the differentiable MPC policy can naturally be adapted to objectives *besides* simple state prediction, such as incorporating the additional cost learning portion.

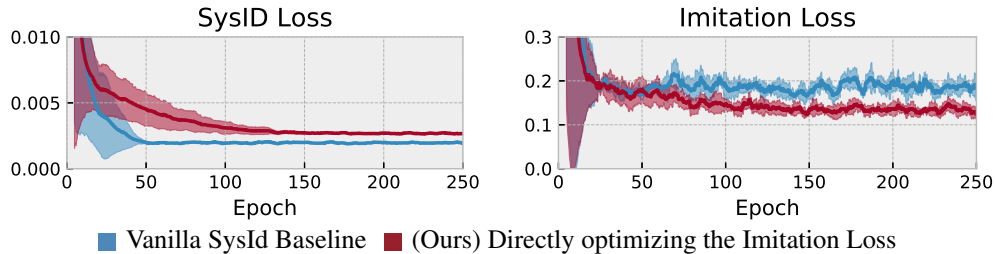

Figure 5: Convergence results in the non-realizable Pendulum task.

## 5.4 Imitation Learning: SysId with a non-realizable expert

All of our previous experiments that involve SysId and learning the dynamics are in the unrealistic case when the expert's dynamics are in the model class being learned. In this experiment we study a case where the expert's dynamics are *outside* of the model class being learned. In this setting we will do imitation learning for the parameters of a dynamics function with vanilla SysId and by directly optimizing the imitation loss (*sysid* and the *mpc.dx* in the previous section, respectively).

SysId often fits observations from a noisy environment to a simpler model. In our setting, we collect optimal trajectories from an expert in the pendulum environment that has an additional damping term and also has another force acting on the point-mass at the end (which can be interpreted as a "wind" force). We do learning with dynamics models that *do not* have these additional terms and therefore we *cannot* recover the expert's parameters. Figure 5 shows that even though vanilla SysId is slightly better at optimizing the next-state transitions, it finds an inferior model for imitation compared to our approach that directly optimizes the imitation loss.

We argue that the goal of doing SysId is rarely in isolation and always serves the purpose of performing a more sophisticated task such as imitation or policy learning. Typically SysId is merely a surrogate for optimizing the task and we claim that the task's loss signal provides useful information to guide the dynamics learning. Our method provides one way of doing this by allowing the task's loss function to be directly differentiated with respect to the dynamics function being learned.

## 6 Conclusion

This paper lays the foundations for differentiating and learning MPC-based controllers within reinforcement learning and imitation learning. Our approach, in contrast to the more traditional strategy of "unrolling" a policy, has the benefit that it is much less computationally and memory intensive, with a backward pass that is essentially free given the number of iterations required for a the iLQR optimizer to converge to a fixed point. We have demonstrated our approach in the context of imitation learning, and have highlighted the potential advantages that the approach brings over generic imitation learning and system identification.

We also emphasize that one of the primary contributions of this paper is to define and set up the framework for differentiating through MPC in general. Given the recent prominence of attempting to incorporate planning and control methods into the loop of deep network architectures, the techniques here offer a method for efficiently integrating MPC policies into such situations, allowing these architectures to make use of a very powerful function class that has proven extremely effective in practice. The future applications of our differentiable MPC method include tuning model parameters to task-specific goals and incorporating joint model-based and policy-based loss functions; and our method can also be extended for stochastic control.

**Acknowledgments**

BA is supported by the National Science Foundation Graduate Research Fellowship Program under Grant No. DGE1252522. We thank Alfredo Canziani, Shane Gu, and Yuval Tassa for insightful discussions.

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
