[Supplementary Material · empc-sup.pdf]

# A  LQR and MPC Algorithms

---

**Algorithm 1** $\mathrm{LQR}_T(x_{\mathrm{init}}; C, c, F, f)$          *Solves Equation (2) as described in Levine [2017]*

---

The **state space** is $n$-dimensional and the **control space** is $m$-dimensional.
$T \in \mathbb{Z}_+$ is the **horizon length**, the number of nominal timesteps to optimize for in the future.
$x_{\mathrm{init}} \in \mathbb{R}^n$ is the initial state
$C \in \mathbb{R}^{T \times n+m \times n+m}$ and $c \in \mathbb{R}^{T \times n+m}$ are the quadratic cost terms. Every $C_t$ must be PSD.
$F \in \mathbb{R}^{T \times n \times n+m}$ $f \in \mathbb{R}^{T \times n}$ are the affine cost terms.

---

  ▷ **Backward Recursion**
  $V_T = v_T = 0$
  **for** t = T to 1 **do**
     $Q_t = C_t + F_t^\top V_{t+1} F_t$
     $q_t = c_t + F_t^\top V_{t+1} f_t + F_t^\top v_{t+1}$
     $K_t = -Q_{t,uu}^{-1} Q_{t,ux}$
     $k_t = -Q_{t,uu}^{-1} q_{t,u}$
     $V_t = Q_{t,xx} + Q_{t,xu} K_t + K_t^\top Q_{t,ux} + K_t^\top Q_{t,uu} K_t$
     $v_t = q_{t,x} + Q_{t,xu} k_t + K_t^\top q_{t,u} + K_t^\top Q_{t,uu} k_t$
  **end for**

  ▷ **Forward Recursion**
  $x_1 = x_{\mathrm{init}}$
  **for** t = 1 to T **do**
     $u_t = K_t x_t + k_t$
     $x_{t+1} = F_t \begin{bmatrix} x_t \\ u_t \end{bmatrix} + f_t$
  **end for**

  **return** $x_{1:T}, u_{1:T}$

---

**Algorithm 2** $\mathrm{MPC}_{T,\underline{u},\overline{u}}(x_{\mathrm{init}}, u_{\mathrm{init}}; C, f)$ *Solves Equation (10) as described in Tassa et al. [2014]*

---

The **state space** is $n$-dimensional and the **control space** is $m$-dimensional.
$T \in \mathbb{Z}_+$ is the **horizon length**, the number of nominal timesteps to optimize for in the future.
$\underline{u}, \overline{u} \in \mathbb{R}^m$ are respectively the control **lower-** and **upper-bounds**.
$x_{\mathrm{init}} \in \mathbb{R}^n, u_{\mathrm{init}} \in \mathbb{R}^{T \times m}$ are respectively the initial state and nominal control sequence
$C : \mathbb{R}^{n \times m} \to \mathbb{R}$ is the non-convex and twice-differentiable **cost function**.
$F : \mathbb{R}^{n \times m} \to \mathbb{R}^n$ is the non-convex and once-differentiable **dynamics function**.

---

$x_1^1 = x_{\mathrm{init}}$
**for** t = 1 to T-1 **do**
$\quad x_{t+1}^1 = f(x_t, u_{\mathrm{init},t})$
**end for**
$\tau^1 = [x^1, u_{\mathrm{init}}]$

**for** i = 1 to *[converged]* **do**
$\quad$ **for** t = 1 to T **do**
$\quad\quad \triangleright$ Form the *second-order Taylor expansion* of the cost as in Equation (12)
$\quad\quad C_t^i = \nabla_{\tau_t^i}^2 C(\tau_t^i)$
$\quad\quad c_t^i = \nabla_{\tau_t^i} C(\tau_t^i) - (C_t^i)^\top \tau_t^i$

$\quad\quad \triangleright$ Form the *first-order Taylor expansion* of the dynamics as in Equation (13)
$\quad\quad F_t^i = \nabla_{\tau_t^i} f(\tau_t^i)$
$\quad\quad f_t^i = f(\tau_t^i) - F_t^i \tau_t^i$
$\quad$ **end for**
$\quad \tau_{1:T}^{i+1} = \mathrm{MPCstep}_{T,\underline{u},\overline{u}}(x_{\mathrm{init}}, C, f, \tau_{1:T}^i, C^i, c^i, F^i, f^i)$
**end for**

**function** $\mathrm{MPCstep}_{T,\underline{u},\overline{u}}(x_{\mathrm{init}}, C, f, \tau_{1:T}, \tilde{C}, \tilde{c}, \tilde{F}, \tilde{f})$
$\quad \triangleright C, f$ are the *true cost* and *dynamics* functions. $\tau_{1:T}$ is the *current trajectory* iterate.
$\quad \triangleright \tilde{C}, \tilde{c}, \tilde{F}, \tilde{f}$ are the *approximate cost* and *dynamics* terms around the current trajectory.

$\quad \triangleright$ **Backward Recursion:** Over the linearized trajectory.
$\quad V_T = v_T = 0$
$\quad$ **for** t = T to 1 **do**
$\quad\quad Q_t = \tilde{C}_t + \tilde{F}_t^\top V_{t+1} \tilde{F}_t$
$\quad\quad q_t = \tilde{c}_t + \tilde{F}_t^\top V_{t+1} \tilde{f}_t + \tilde{F}_t^\top v_{t+1}$

$\quad\quad k_t = \mathrm{argmin}_{\delta u} \; \frac{1}{2} \delta u^\top Q_{t,uu} \delta u + Q_x^\top \delta u \; \text{ s.t. } \; \underline{u} \le u + \delta u \le \overline{u}$
$\quad\quad \triangleright$ Can be solved with a *Projected-Newton method* as described in Tassa et al. [2014].
$\quad\quad \triangleright$ Let $f, c$ respectively index the *free* and *clamped* dimensions of this optimization problem.

$\quad\quad K_{t,f} = -Q_{t,uu,f}^{-1} Q_{t,ux}$
$\quad\quad K_{t,c} = 0$

$\quad\quad V_t = Q_{t,xx} + Q_{t,xu} K_t + K_t^\top Q_{t,ux} + K_t^\top Q_{t,uu} K_t$
$\quad\quad v_t = q_{t,x} + Q_{t,xu} k_t + K_t^\top q_{t,u} + K_t^\top Q_{t,uu} k_t$
$\quad$ **end for**

$\quad \triangleright$ **Forward Recursion and Line Search:** Over the true cost and dynamics.
$\quad$ **repeat**
$\quad\quad \hat{x}_1 = \tau_{x_1}$
$\quad\quad$ **for** t = 1 to T **do**
$\quad\quad\quad \hat{u}_t = \tau_{u_t} + \alpha k_t + K_t(\hat{x}_t - \tau_{x_t})$
$\quad\quad\quad \hat{x}_{t+1} = f(\hat{x}_t, \hat{u}_t)$
$\quad\quad$ **end for**
$\quad\quad \alpha = \gamma \alpha$
$\quad$ **until** $\sum_t C([\hat{x}_t, \hat{u}_t]) \le \sum_t C(\tau_t)$

$\quad$ **return** $\hat{x}_{1:T}, \hat{u}_{1:T}$
**end function**

---

# B   Imitation learning experiment losses

Figure 6: Learning results on the (simple) pendulum and cartpole environments. We select the best validation loss observed during the training run and report the corresponding train and test loss. Every datapoint is averaged over four trials.