[Reviews · NeurIPS 2018]

Reviewer 1



The paper presents a technique of making the MPC cost function differentiable to enable end to end learning. The MPC cost is analytically differentiated through convex approximation at a fixed point using the KKT condition. The MPC cost is formed as a box constrained quadratic program and the system is solved using an LQR solver and the derivatives are computed based on the OptNet approach. The advantage of the approach is that the backward pass is only computed once instead of the normal unrolling strategy used in the past literature. The approach is tested for linear dynamical systems and in continuos control domains like pendulum, cartpole and car model setting and is compared with other imitation learning approaches. The paper applies the method of solving quadratic program using efficient differentiable solver introduced in [1] to the MPC cost function which makes the novelty in the paper very limited. There are various limitations of these type of solvers like high memory complexity of the solver which makes it difficult to scale to networks with large number of parameters and also as discussed in the paper, its only applicable to systems where the controller reaches a fixed point which makes it limited in the applications it can be used. Minor Correction: The formatting of the references is incorrect. [1] Brandon Amos and J Zico Kolter. OptNet: Differentiable optimization as a layer in neural networks. In International Conference on Machine Learning, pages 136–145, 2017. Update: I have changed my score to a borderline accept after going through the author's feedback and other reviewers' comments. I agree with the authors that the work lays the foundation for differentiable control but is missing discussions on effectiveness of the approach in real world settings which the author have agreed to address in the revised version.

Reviewer 2



The paper presents an approach, from an optimization perspective, to integrate both the controller and the model. I am glad to see that this work has abundant citations from both classical control theory community and reinforcement learning community. And the idea of formulating the problem is nice, however I have concerns about clarity and significance. First, the formulation of the problem does not take into account any randomness (those from environment) and does not even mention Markov decision process. In the equation 2, there is no expectation operator available. At least for people from reinforcement learning community, the clarity can be improved if you start the formulation from some expected performance measure (return). With the current formulation and proposed method, I am doubting the applicability of this method. The paper emphasized some previous work requires an "unrolling" process for optimization, however, I think those works are used for other purposes instead of imitation learning? The motivation of this work should be better addressed. Second, although it is fine to use the proposed method for imitation learning, given the comparison criterion imitation loss and model loss, why not compare with a naive method such as learn the model and imitation loss simultaneously from expert demonstrations? Importantly, the paper should clearly state what are available in the imitation learning setting ( the expert itself available, or only demonstrations, or sth else)? From the paper, I think there is a somewhat strong assumption that the expert itself should be available, and I cannot intuitively think of how the proposed method can learn a policy which is better than the experts'. Third, the paper claims the proposed method has a lower computation and space complexity. It should be better to clearly identify the complexity in asymptotic notation (per iteration). Or at least add some explanation about why it can save space complexity. Last, the paper should include an appendix, with more implementation and experimental details. Update: the response addresses part of my concern. I upgraded my rating. I am still concerning about its applicability.

Reviewer 3



\documentclass[12pt,a4paper]{article} \usepackage[utf8]{inputenc} \usepackage{amsmath} \usepackage{amsfonts} \usepackage{amssymb} \begin{document}  Consider rendering this review in LaTeX for a better experience.   \section{Paper summary}  This paper builds upon the recently proposed approach~\cite{amos2017optnet} to differentiating through quadratic programs (QP). More precisely, if $x^\star$ is the solution of a QP, the algorithm developed in~\cite{amos2017optnet} provides an efficient way to find derivatives of a scalar loss function $l(x^\star)$ with respect to the parameters of the QP, i.e., the matrices and vectors in the definition of the QP.  Since the linear quadratic regulator (LQR) problem is but a large QP, one can apply the methodology from~\cite{amos2017optnet}. Due to the chain structure of the computation graph for the LQR, the problem of finding derivatives of a loss function defined over optimal trajectories can be efficiently solved using Riccati recursion, which is summarized in Algorithm~1.  A non-trivial insight of the paper is to extend this approach to non-linear trajectory optimization problems. Popular sequential quadratic programming (SQP) solvers (including iLQR) proceed by solving a sequence of local QP approximations of the original non-linear trajectory optimization problem. Consequently, once converged, they output the `optimal' quadratic approximation of the original problem as a by-product. By applying Algorithm~1 to this surrogate QP, one can find the exact derivatives of a loss function defined on an optimal trajectory.  Another contribution of the paper is to incorporate box constraints on the controls by considering box-constrained QPs. The full procedure of differentiating through non-linear trajectory optimization is described in Algorithm~2.  The main advantage of the proposed algorithm is in providing constant-time backward pass for evaluating the derivative. Prior works would unroll all iterations of the SQP solver, thus the backward pass would scale linearly with the number of SQP iterations. Figure~1 shows the dramatic effect of the simplification that the proposed algorithm provides.  As a proof of concept, the authors apply Algorithm~1 to LQR system identification problem with time-invariant matrices $A$ and $B$. The unusual feature of the approach taken in this paper is that system identification happens as a consequence of minimizing quadratic imitation loss between actions of the optimal controller and the one with adjustable matrices $A$ and $B$. This experiment serves mainly to confirm that the equations derived in the paper are correct because, of course, this is not the most efficient way to do system identification.  Another set of simulated toy tasks shows that the algorithm can recover the system dynamics as well as the reward function from observed sequences of optimal controls $u_{1:T}(x; \theta)$ resulting from an initial state $x$.   \section{Paper review} Differentiation through LQR presented in this paper is a sound approach. However, I think a bit more care is needed in the non-linear trajectory optimization case. This point is explained in the following subsections.  \subsection{Title is misleading} The title reads ``differential MPC'', but the presented algorithm is actually ``differentiable non-linear trajectory optimization''. The controllers considered in the paper are of the form $u^\star_{1:T} = \pi(x_1; f_\theta, c_\theta)$ where $f_\theta(x,u)$ denotes the model of the dynamics and $c_\theta(x,u)$ denotes the model of the cost function. Here, $\pi$ is a function performing non-linear trajectory optimization. Derivatives that the authors propose are taken with respect to the optimal trajectory $\tau_{1:T} = (x^\star, u^\star)_{1:T}$.  On the other hand, an MPC controller has the form $u^\star_1 = \pi(x_1; f_\theta,c_\theta)$. So, only the first action returned by the trajectory optimizer gets executed. Then, the next state of the system is $x' = f(x, \pi(x; f_\theta,c_\theta))$. Differentiating a function of a trajectory generated in such a way with respect to $\theta$ is much more involved. But this is what one would expect to be called ``differentiable MPC''.  \subsection{LQR with box constraints} In Line 170, the authors write ``we use an LQR solver that zeros the appropriate controls''. What exactly is meant by zeroing appropriate controls? As far as I can tell, Riccati recursion does not work if one has box constraints. Instead, one has to solve a box-constrained QP at every time step  of a trajectory, starting at the end. But that sounds like a lot of computation. Is that how it is done? Or do you compute an unconstrained LQR and then clamp the controls?  \subsection{Local minima in non-linear optimization} How do you deal with the problem that the SQP solver delivers only a local optimum? The solution to which the solver converges in the forward run is usually highly dependent on the initial trajectory. That means the derivative computed according to the proposed method depends on the initialization of the SQP solver.  \subsection{Questions about experiments} In Line 234, the authors write ``we use LQR experts''. Do you mean `iLQR experts'? Pendulum and cartpole are nonlinear problems, so I assume `iLQR' is meant here.  Could you elaborate on the cost functions in your experiments and whether you use box constraints on the controls, please? More concretely, I am wondering if the pendulum is underpowered in your experiments.  What kind of model are you learning? Is it time-varying or time-invariant?  A partial explanation of why the experiments where both the dynamics and the cost function are learned are not as conclusive may lie in the non-uniqueness of the solution: there may be more than one combination of dynamics and cost that yield the same controls. As an example, consider 1-step LQR with 1D state and controls. The total cost is \begin{equation}   J = Qx_1^2 + Ru_1^2 + Qx_2^2. \end{equation} Substitute $x_2 = ax_1 + bu_1$ to get \begin{equation}   J = (Q + Qa^2)x_1^2 + (R + Qb^2) u_1^2 + 2 Q ab x_1 u_1. \end{equation} Optimizing it with respect to $u_1$, one finds \begin{equation}   u_1 = \frac{Qab}{R+Qb^2} x_1. \end{equation} From here, it is clear that if we allow both $\{a,b\}$ and $\{Q,R\}$ to vary, there is more than one combination of them that yields the same gain.   \subsection{Minor remarks} 1) Why are plots cut at $1\mathrm{e}3$ iterations? 2) Why is the car going left in the beginning in Figure 3? 3) Why is SysID cost oscillating in Figure 4? 4) Somwhere in the text, you call derivatives `analytic'. I think calling them `exact' would be more appropriate, since there is no closed-form formula, only an algorithm for computing them.   \section{Update after rebuttal} I would like to thank the authors for addressing my comments. In my opinion, the paper provides a solid contribution and I vote for accepting it.  \begin{thebibliography}{9}  \bibitem{amos2017optnet} Amos, B., \& Kolter, J. Z. (2017). Optnet: Differentiable optimization as a layer in neural networks. \textit{arXiv preprint arXiv:1703.00443.}   \end{thebibliography}   \end{document}